# OpenReview forum: "Output Supervision Can Obfuscate the Chain of Thought"
_ICLR.cc/2026/Conference — ICLR 2026 Poster_

### Official Review · Reviewer_Nrs7 · 2025-10-30

**Soundness:** 3
**Presentation:** 3
**Contribution:** 3
**Rating:** 6
**Confidence:** 4

**Summary:**

This work discusses an interesting phenomenon; supervising the model outputs in order to prevent specific properties appearing in them, causes a "spillover" effect where the same properties are also removed (i.e., obfuscated) from the chain-of-thought (CoT). The authors attribute this effect to two separate mechanisms, parametric feedback and conditional feedback, obtained via analyzing the structure of the policy gradient on a loss that involves the task and an output obfuscation loss. The first is due to the parameter configurations leading to output obfuscation also generalizing to CoT obfuscation whereas the second is due to the observation that obfuscated CoTs might be needed in order to produce obfuscated outputs. Based on the above observations, the authors propose a specific remedy for each mechanism; they zero-out a specific gradient term in order to prevent conditional feedback and use separate models for the CoT and output in order to prevent the parametric feedback. Finally, they demonstrate on three separate settings that feedback spillover occurs and show how the proposed solutions can help prevent it.

**Strengths:**

- This work highlights a novel observation, feedback spillover, that manifests when doing output supervision, the suggested way to avoid CoT obfuscation when training against monitors. This is a timely contribution to the emerging field of CoT monitoring.
- The experimental setups to measure CoT obfuscation as well as the presentation of feedback spillover and the methods to tackle it are clear and easy to follow.
- The insights and recommendations to practitioners are valuable and actionable.

**Weaknesses:**

- The solutions provided by the authors feel like a "band-aid"; they just push the spillover contributing terms of the loss gradient to zero, which might negatively affect the model performance and overall optimization of the specific loss employed (e.g., what is the impact of the "mind & face" strategy on just the task reward? Is it lower than the standard one where the same model is used for the CoT and answer?). The contribution of this work would have been stronger if the authors instead articulated what loss functions might be better / do not lead to feedback spillover when doing output supervision.
- Even though the authors clearly define parametric and conditional feedback spillover, they do not measure it in the experiments. This work can be improved by showing what kind of spillover exists in each of the experimental setups, thus providing extra motivation / explanation for why, e.g., "reward targeting" or "mind & face" might work better.

**Questions:**

Overall, I am leaning on the positive side for this work due to the useful insights it provided. Having said that, addressing the weaknesses and comments below will further convince me.

- How does feedback spillover manifest when the CoT is not necessary for better downstream task performance (i.e., cases where the model does not need to put critical information in it)? Intuitively, if one puts pressure on specific structures for the output this would also force the CoT to change when the model depends strongly on CoT for solving a task.
- How does the spillover effect change during training in the simple case of "Task reward + output monitor" without "reward targeting" and "Mind & Face"? Does it emerge from the get-go or at the end of training? This would highlight whether the spillover could be solved with other means, e.g., regularization or early stopping.
- What were the architectures for the Mind and Face at each of the experimental settings? Was it the same model at different checkpoints or a completely different model altogether? Did the Mind and Face have similar model capacity? This is important information that needs to be in the main text as it can highlight what practical limitations this strategy has, especially given that it requires having access to two separate models.
- Minor typo: in equation 21 in the appendix there is a missing dependence on $\theta$ on the right hand side.

---

> ### Author Response · Authors · 2025-11-26
>
> Hello, and thank you for taking the time to give feedback on our paper - we’re encouraged by the positive review. Below, we respond to each of the weaknesses and questions raised.
>
>
> **Weaknesses**
>
> > W1:  The solutions provided by the authors feel like a "band-aid"; they just push the spillover contributing terms of the loss gradient to zero, which might negatively affect the model performance and overall optimization of the specific loss employed (e.g., what is the impact of the "mind & face" strategy on just the task reward? Is it lower than the standard one where the same model is used for the CoT and answer?). The contribution of this work would have been stronger if the authors instead articulated what loss functions might be better / do not lead to feedback spillover when doing output supervision.
>
> The task rewards for each type of run, including Mind & Face, are available in our results tables. The only environment where Mind & Face had significantly worse task reward than the standard Penalty run was the MMLU-with-hints setting. Future work may further investigate the effect of Mind & Face on performance; for the purposes of the current work, we think it is sufficient to demonstrate a Pareto improvement in terms of Training Reward and CoT Monitor Detection Rate.
>
>
> > W2:  Even though the authors clearly define parametric and conditional feedback spillover, they do not measure it in the experiments. This work can be improved by showing what kind of spillover exists in each of the experimental setups, thus providing extra motivation / explanation for why, e.g., "reward targeting" or "mind & face" might work better.
>
> We agree that further analysis could be informative here—for example, one could track the norms of the gradient terms corresponding to conditional and parametric spillover throughout training—but we leave this for follow-up work since we don’t see it as essential to the main story of the paper.
>
>
> **Questions**
>
>
> > Q1:  How does feedback spillover manifest when the CoT is not necessary for better downstream task performance (i.e., cases where the model does not need to put critical information in it)? Intuitively, if one puts pressure on specific structures for the output this would also force the CoT to change when the model depends strongly on CoT for solving a task.
>
> This is a good question, and we believe the answer is already contained in the paper. E.g. in the MMLU-with-hints task, the model does not need its CoT to solve the task, whereas in the polynomial-factoring environment, the model needs its CoT. The task rewards, training rewards and output penalties are given in our Results section for both these environments.
>
> > Q2:  How does the spillover effect change during training in the simple case of "Task reward + output monitor" without "reward targeting" and "Mind & Face"? Does it emerge from the get-go or at the end of training? This would highlight whether the spillover could be solved with other means, e.g., regularization or early stopping.
>
> We have added Appendix D, where we provide training curves for each environment.
>
>
> > Q3:  What were the architectures for the Mind and Face at each of the experimental settings? Was it the same model at different checkpoints or a completely different model altogether? Did the Mind and Face have similar model capacity? This is important information that needs to be in the main text as it can highlight what practical limitations this strategy has, especially given that it requires having access to two separate models.
>
> We added a line (line 261-262) explaining that, although in principle one could use different models for the Mind and Face, in all our experiments they were initialized as copies of the same model.
>
> We thank the reviewer again for their feedback.

---

### Official Review · Reviewer_dHe4 · 2025-10-31

**Soundness:** 4
**Presentation:** 3
**Contribution:** 4
**Rating:** 8
**Confidence:** 4

**Summary:**

- This paper demonstrates that supervision of LLM outputs can “spillover” to increase rates of obfuscated CoT, demonstrating this effect in multiple environments.
- They identify two mechanisms by which this can occur: parametric feedback spillover and conditional feedback spillover, and provide a theoretical analysis of these phenomena.
- This theoretical analysis serves to motivate two corresponding mitigations. 'Reward targeting' mitigates conditional feedback spillover by preventing gradients from the outputs flowing through CoT tokens. 'Mind and face' uses separate models to generate the CoT and output, respectively, thus preventing sharing gradients over the same parameters.
- They show these mitigations can, to some extent, reduce feedback spillover, though results are varied and often task performance is impacted.
- The paper ends with important recommendations to developers, based on their findings

**Strengths:**

- The demonstration of feedback spillover seems original and significant. CoT monitorability is increasingly highlighted as a promising and important approach for safety. The existence and strength of the feedback spillover effect here is surprising. Naive safety approaches may not have otherwise considered the possibility of this effect. Developers should indeed take note of these results and follow the provided recommendations. I am excited to see follow-up work further investigating this phenomenon and improving corresponding mitigations.
- The paper provides a clear theoretical analysis that explains the potential mechanisms behind the phenomenon. This is a valuable contribution to our understanding of feedback spillover.
- The initial attempts at mitigating feedback spillover are well-motivated, based on the prior theoretical analysis. Whilst the mitigations are not perfect, these are valuable initial attempts and they highlight the difficulty of mitigating this effect.

**Weaknesses:**

The settings are narrow and somewhat toyish.

There is a lack of focus on the extent to which the models reasoning is “load-bearing” - does the model require its reasoning to optimize task reward in any of these settings? This seems an important distinction to be aware of when interpreting the results. It would be valuable to compare results in environments where reasoning is and isn’t important for the task reward. Relatedly, do the authors think the reward targeting mitigation would significantly hurt task reward in settings where reasoning is required?

While the theoretical analysis of feedback spillover is strong, it could be nice to see more results that provide evidence and a more detailed, empirically-grounded understanding of the two spillover mechanisms. The mitigation results do seem to bear on this, but further depth of analysis could be provided.

There is not many details provided regarding why these particular models and sizes were chosen. Only Qwen-3 models are used, and it is unclear how specific these results are to these models, and to these smaller sizes.

**Questions:**

- I sometimes had some conceptual confusions when reading the results section. I was often unclear how exactly the experimental setup could map to a realistic safety-related scenario (e.g., in the maths setting, should we think of “expanding the derivative” as a misaligned action that is incentivized by a misspecified reward?). I was initially unclear whether the mitigations should seek to maintain a high task reward on the pareto frontier - the high-task reward corresponds to an “undesired” action - this could be clarified.
- In the maths environment it seems the model learns to perform the math in an entirely different way, rather than obfuscating its initial approach? This seems a notable distinction from “obfuscation” of reasoning?
- I found it confusing when the “Training Reward” metric included the output penalty for “no penalty” training runs which did not use the penalty in their training reward.
- To what extent did you optimize the experimental settings you used to achieve these spillover results? Did you try other environments where no spillover was achieved?
- Could you provide basic details of your RL setup in the main paper? (e.g., what algorithm did you use?)
- Could you provide training curves? It would be interesting to see the initialization points of the various reward terms, and how the rewards evolve over time.

---

> ### Author Response · Authors · 2025-11-26
>
> We really appreciate the reviewer’s in-depth feedback, and their positivity about our work. Below, we respond to each of the weaknesses and questions raised
>
> **Weaknesses**
>
> > W1: The settings are narrow and somewhat toyish.
>
> The settings are indeed simple. We had limited compute and wanted to provide a clean demonstration of spillover. That said, the settings were not chosen arbitrarily: they were either taken from, or directly inspired by, major recent papers on CoT faithfulness/obfuscation from each leading AI lab:
>
> - Polynomial factoring is from [GDM 2025](https://arxiv.org/abs/2507.05246)
> - Questions-with-hints is from [Anthropic 2025](https://arxiv.org/abs/2505.05410)
> - Our terminal-based reward hacking environment is inspired by the coding environment in [OpenAI 2025](https://arxiv.org/abs/2503.11926) (though simplified so that mid-sized models can perform well)
>
> We think that demonstrating feedback spillover in these pre-existing environments provides assurance that the settings were not cherry-picked.
>
>
> > W2:  There is a lack of focus on the extent to which the models reasoning is “load-bearing” - does the model require its reasoning to optimize task reward in any of these settings?
>
> The model requires CoT to complete the polynomial factoring task, but not the other two tasks. We added a brief note (lines 274-276) to highlight this in the paper.
>
>
> > W3:  Relatedly, do the authors think the reward targeting mitigation would significantly hurt task reward in settings where reasoning is required?
>
> In Reward Targeting, the task reward is still used to train the CoT, as usual. It is only the monitor penalty that is not used to train the CoT. Therefore we do not expect Reward Targeting to significantly hurt task reward. And indeed, Reward Targeting did not significantly decrease task reward relative to the Penalty runs in any of our environments (even in the polynomial factoring environment, where CoT was necessary).
>
>
> > W4:  While the theoretical analysis of feedback spillover is strong, it could be nice to see more results that provide evidence and a more detailed, empirically-grounded understanding of the two spillover mechanisms.
>
> We agree that further analysis could be informative here—for example, one could track the norms of the gradient terms corresponding to conditional and parametric spillover throughout training—but we leave this for follow-up work since we don’t see it as essential to the main story of the paper.
>
>
> > W5:  There is not many details provided regarding why these particular models and sizes were chosen. Only Qwen-3 models are used, and it is unclear how specific these results are to these models, and to these smaller sizes.
>
> Qwen3 was chosen due to being one of the performant suites of open-source reasoning models at the time of writing. We agree that it would be nice to confirm our results for larger models and different model families, but we choose to leave this for future work.
>
> [continued in next comment]

---

> ### Author Response · Authors · 2025-11-26
>
> [continued from previous comment]
>
> **Questions**
>
> > Q1:  I sometimes had some conceptual confusions when reading the results section. I was often unclear how exactly the experimental setup could map to a realistic safety-related scenario (e.g., in the maths setting, should we think of “expanding the derivative” as a misaligned action that is incentivized by a misspecified reward?). I was initially unclear whether the mitigations should seek to maintain a high task reward on the pareto frontier... this could be clarified.
>
> We agree that mapping the polynomial environment onto a safety-related scenario is rather confusing. We have added a sentence (lines 275-9) clarifying that, whilst we do observe feedback spillover in this environment, we do not say that obfuscation has occurred. This sentence also explicitly states that we interpret “expanding the derivative” as the bad behavior. We also clarify that in all cases, we’d like task reward to be high (lines 258-9).
>
>
> > Q2:  In the maths environment it seems the model learns to perform the math in an entirely different way, rather than obfuscating its initial approach? This seems a notable distinction from “obfuscation” of reasoning?
>
> (Addressed in our answer to Q1)
>
>
> > Q3:  I found it confusing when the “Training Reward” metric included the output penalty for “no penalty” training runs which did not use the penalty in their training reward.
>
> We plotted Training Reward to enable a fair comparison between runs. We imagine that high Training Reward is the objective of the developer, since they want 1) high task reward and 2) no bad behavior detected in the output.
>
>
> > Q4:  To what extent did you optimize the experimental settings you used to achieve these spillover results? Did you try other environments where no spillover was achieved?
>
> In some experimental settings (not reported in the paper) we were unable to induce output obfuscation whatsoever. But whenever output obfuscation did occur in a given environment, we saw spillover in that environment.
>
>
> > Q5:  Could you provide basic details of your RL setup in the main paper? (e.g., what algorithm did you use?)
>
> We added lines 260-61 explaining that we used the REINFORCE algorithm, with hyperparameters as given in Appendix C.
>
>
> > Q6:  Could you provide training curves? It would be interesting to see the initialization points of the various reward terms, and how the rewards evolve over time.
>
> We added Appendix D, where we provide training curves for each environment.
>
> Thanks again for the feedback!

---

### Official Review · Reviewer_VZAn · 2025-11-01

**Soundness:** 3
**Presentation:** 3
**Contribution:** 3
**Rating:** 6
**Confidence:** 2

**Summary:**

The paper introduces the problem of feedback spillover when training LLMs with output-only supervision, especially modern large models trained with RL. It shows a gap between standard LLM evaluation metrics and genuine reasoning ability, which can limit the interpretability and safety of using CoTs to understand LLMs, undermining the monitorability of CoT.

Additionally, the authors introduce two mitigation methods for feedback spillover, "Mind & Face" Models and Reward Targeting.

**Strengths:**

- Introduces the core concept of feedback spillover, which can cause Obfuscated CoT,

- Monitoring the logical soundness of the CoT process as opposed to output only during training is quite novel.

- Empirically showed that OS-Models have higher accuracy than CoT-Models, but suffer from invalid CoT monitoring.

**Weaknesses:**

- The "Mind & Face" model using disjoint parameters to avoid obfuscation does make sense; however, I'm concerned on the computation expense and the possibility of error accumulation, which could harm accuracy or lose some of the supervision signals during training from either the CoT process or the outputs.

**Questions:**

- Feedback spillover is defined for RL paradigm, but does the same formulation still hold for supervised fine-tuning?

- Does this phenomenon occur more for even larger base models?

- The paper measures spillover through lower CoT monitor scores. To what extent can this reflect safer CoT learning / what are the limitations of this scoring method?

---

> ### Author Response · Authors · 2025-11-26
>
> We appreciate the reviewer’s feedback and thank them for their time. We are glad you found the introduction of feedback spillover a strength of our paper.
>
> **Weaknesses**
>
> > The "Mind & Face" model using disjoint parameters to avoid obfuscation does make sense; however, I'm concerned on the computation expense and the possibility of error accumulation, which could harm accuracy or lose some of the supervision signals during training from either the CoT process or the outputs.
>
> We acknowledge that Mind & Face introduces significant additional computational cost. In lines 452-458, we offer some suggestions that could reduce this cost. Reducing the computational cost of Mind & Face is important future work, and we hope the suggestions in our paper are useful to future work that attempts this. In Section 5, we also make the suggestion that Mind & Face could be useful for reasons other than improving CoT monitorability. For example, a model developer could swap out different Face models to access models with different personas. This may help justify the added cost of Mind & Face. Additionally, our other mitigation, reward targeting, comes at no additional computation cost.
>
> Regarding the possibility of error accumulation, we found that Mind & Face did not negatively impact performance in the polynomial derivative factoring environment or multi-turn terminal interaction environment. We did observe a performance degradation for all methods in the QA with hints environment, although we also saw that our mitigations could increase the training reward, such as in Figure 7.
>
> **Questions**
>
> > Feedback spillover is defined for RL paradigm, but does the same formulation still hold for supervised fine-tuning?
>
> While we believe non-RL settings are important to study for safety, we suggest that feedback spillover is best studied in the RL setting. This is because in supervised fine-tuning, the CoT the model should output for a given sample is already provided as a supervised example, leaving the more direct mitigation of providing monitorable CoTs as supervised examples, rather than more complicated mitigations such as the ones in our paper. Further, CoT models are predominantly trained with RL, which makes it natural to study CoT monitorability in the RL setting.
>
> > Does this phenomenon occur more for even larger base models?
>
> In the polynomial derivative factoring environment we give results for Qwen3-14B, which is the largest model we experiment with. We suggest that this is quite large for an open source model, but would hope that future work studies feedback spillover in larger models, as we acknowledge that this is an important open question.
>
> > The paper measures spillover through lower CoT monitor scores. To what extent can this reflect safer CoT learning / what are the limitations of this scoring method?
>
> CoT monitor scores will reflect the safety of the CoT according to the capabilities and goal of the monitor. In our paper, we try to study feedback spillover in a way that focus on CoT monitorability as a whole, rather than the objective of a specific monitor. If our monitors weren’t reflective of the way CoT monitors are usually implemented, then that could be a limitation of our scoring method. However, our paper features multiple CoT monitors based on LLM judges, which is a standard design pattern for CoT monitors. We believe that our CoT monitors are similar enough to CoT monitors that could be used in production that our results are informative for applications in the real world.

---

### Official Review · Reviewer_DW5q · 2025-11-01

**Soundness:** 3
**Presentation:** 3
**Contribution:** 3
**Rating:** 8
**Confidence:** 4

**Summary:**

This paper investigates feedback spillover, a phenomenon where training a model against an output-only monitor (rather than a CoT monitor) still leads to obfuscated chains of thought (CoTs). The authors argue that output supervision can indirectly shape CoTs through two mechanisms:

* Parametric spillover: Parameter updates that make outputs look safe generalize to CoTs.
* Conditional spillover: Safe-looking CoTs increase the likelihood of safe outputs, reinforcing obfuscation.

They propose two mitigations:

1. Mind & Face: Separate models for CoT ("mind") and output ("face").
2. Reward targeting: Modify gradients to prevent reinforcement of CoTs that lead to safe outputs.

Experiments in three RL environments (polynomial derivative factoring, QA with hints, multi-turn terminal interaction) show strong spillover under output penalties and partial success of mitigations. The paper provides a formal gradient decomposition, empirical demonstrations, and practical recommendations for developers.

**Strengths:**

* **Novel insight**: Challenges the assumption that output-only monitoring preserves CoT monitorability. This is a significant contribution to alignment and oversight research.
* **Clear theoretical framing**: The gradient decomposition and formal conditions for spillover (Eq. 5–7) are well-motivated and connect intuitively to the observed behaviors.
* **Mitigation proposals**: Both strategies are conceptually simple yet grounded in the analysis. Reward targeting is especially appealing for its low computational overhead.
* **Empirical breadth**: Demonstrations across diverse environments (math, QA, agentic tasks) strengthen external validity.
* **Practical relevance**: Recommendations for developers and discussion of safety cases make the work actionable.

**Weaknesses:**

* **Mitigation trade-offs**: Reward targeting and Mind & Face often reduce spillover but can destabilize training or hurt task performance (e.g., QA with hints). This limits their practicality without further tuning.
* **Limited model diversity**: Experiments use Qwen variants only; results may not generalize to other architectures or larger frontier models.
* **Hyperparameter sensitivity**: Authors acknowledge instability but don’t provide systematic tuning or ablation studies, leaving uncertainty about robustness.
* **Computational overhead**: Mind & Face doubles model instances, which is costly for large-scale training. Efficiency suggestions are speculative.
* **Scaling and persistence**: Experiments use relatively short training schedules, short trajectories, and bounded CoT lengths; it remains unclear whether feedback spillover's magnitude and stability persist or amplify under longer training and longer-horizon agentic tasks.

**Questions:**

1. Mitigation stability and trade-offs: What factors drive instability when combining Reward Targeting and Mind & Face, and did you try tuning or curriculum strategies to reduce performance drops?



2. Model diversity: Do you expect feedback spillover to generalize beyond Qwen models? Any evidence or theoretical reasons for similar effects in Claude/GPT?



3. Scaling behavior: How does spillover behave under longer training schedules or larger models? Could the effect amplify or diminish with scale?



4. Mitigation efficiency: Mind & Face doubles compute. Did you explore parameter sharing or smaller “face” models to reduce cost without losing effectiveness?


5. Reward targeting generalization: Does reward targeting apply to other oversight signals (toxicity, bias), or is it specific to CoT monitorability?

---

> ### Author Response · Authors · 2025-11-26
>
> We thank the reviewer for their time and expertise. We appreciate that you found our work practically relevant and that the core insight of our paper was novel and clearly theoretically framed.
>
> **Weaknesses**
>
> > Mitigation trade-offs: Reward targeting and Mind & Face often reduce spillover but can destabilize training or hurt task performance (e.g., QA with hints). This limits their practicality without further tuning.
>
> Although in QA with hints simultaneously applying both Mind & Face and reward targeting hurt task performance, the individual techniques had a much smaller effect on task performance (e.g., in Figure 3, the reward targeting run had a better training reward than the run with no mitigations). We also point to the other two environments, in which the training reward of runs with both techniques applied were unaffected, and in some cases superior (e.g., in Figure 7, the Targeted Mind & Face run is significantly more performant than the no mitigation run).
>
> > Hyperparameter sensitivity: Authors acknowledge instability but don’t provide systematic tuning or ablation studies, leaving uncertainty about robustness.
>
> We suggest that in individually studying Mind & Face and reward targeting, we have done the necessary ablations for our mitigations. In both cases we have measured the effects of leaving out the mitigation for all environments. We would appreciate the reviewer elaborating on the hyperparameters they would like to see tuning of.
>
> **Questions**
>
> > Mitigation stability and trade-offs: What factors drive instability when combining Reward Targeting and Mind & Face, and did you try tuning or curriculum strategies to reduce performance drops?
>
> We did not attempt curriculum-based strategies, and would appreciate an elaboration from the reviewer on what they think might be helpful here. Regarding instability, we expect that this comes from how different the Targeted Mind & Face gradient update is from the gradient update with no mitigations. For example, it is possible that there is useful information in the parts of the gradient update we ablate with reward targeting.
>
> > Model diversity: Do you expect feedback spillover to generalize beyond Qwen models? Any evidence or theoretical reasons for similar effects in Claude/GPT?
>
> Our theoretical arguments apply as much to other models as they do to the Qwen models from our paper, leading us to expect to observe feedback spillover in other models also. We discuss the policy gradient in general, making our theoretical arguments quite universal.
>
> > Scaling behavior: How does spillover behave under longer training schedules or larger models? Could the effect amplify or diminish with scale?
>
> In the polynomial derivative factoring environment we give results for Qwen3-14B, which is the largest model we experiment with. As we did not observe a decreasing effect size at models of different sizes in our environments, we do not expect feedback spillover to diminish with scale. Regarding the length of our training runs, we suggest that they are fairly standard for RL post-training, with our longest runs spanning 40,000 episodes in the case of the multi-turn terminal environment.
>
> > Mitigation efficiency: Mind & Face doubles compute. Did you explore parameter sharing or smaller “face” models to reduce cost without losing effectiveness?
>
> We did not explore cost effectiveness strategies for Mind & Face in our paper, although we did offer suggestions in lines 452-458. We acknowledge that Mind & Face could make training and inference significantly more expensive, and methods to reduce its cost are important future work. In Section 5, we also make the suggestion that Mind & Face could be useful for reasons other than improving CoT monitorability. For example, a model developer could swap out different Face models to access models with different personas. This may help justify the added cost of Mind & Face.
>
> > Does reward targeting apply to other oversight signals (toxicity, bias), or is it specific to CoT monitorability?
>
> Although we only experimented with applying reward targeting for improved CoT monitorability, we expect that reward targeting could be useful for other applications also. While we believe this would be out of scope for our paper, we would be excited to see the use of our mitigations for purposes other than CoT monitorability.

---

### Meta-Review · Area_Chair_Z4b7 · 2026-01-06

**Summary:**

This paper studies feedback spillover, a phenomenon in RL-trained language models where output-only supervision (e.g., penalizing unsafe or undesired outputs) unintentionally causes chains-of-thought (CoTs) to become obfuscated, even when CoTs are not directly supervised.

The authors identify two mechanisms behind this effect: Parametric spillover and Conditional spillover. They then provided new mitigation strategies.

**Reviewer Concerns:**

**Experimental Scope and Scaling**

Experiments are seen as somewhat narrow, with only short training horizons, bounded CoT lengths and limited model families.


**Generalization**

Questions about whether feedback spillover applies beyond RL (e.g., supervised fine-tuning) and extends to other oversight signals (toxicity, bias). Some asked for generalizes to frontier models (Claude/GPT)


**Clarity and Experimental Details**

Reward definitions (training reward vs. task reward)
RL setup details and training curves (later addressed in rebuttal)

**Reviewer Scores:**

Reviewer DW5q (Score: 8, accept)
Very positive. Emphasized novelty, theoretical clarity, and real-world relevance. Main concerns were scalability and mitigation trade-offs.

Reviewer dHe4 (Score: 8, accept)
Very positive. Highlighted importance for CoT monitorability and developer practice. Raised thoughtful conceptual questions, but none blocking.

Reviewer Nrs7 (Score: 6, borderline accept)
Generally positive but skeptical of mitigations being “band-aids.” Wanted better loss design and more direct measurement of spillover mechanisms.

Reviewer VZAn (Score: 6, borderline accept)
Positive overall, with concerns about compute cost, larger models, and CoT monitor scoring fidelity.

Clear accept

---

### Decision · Program_Chairs · 2026-01-26

Accept (Poster)